# Targeting Transcription Factors ATF5, CEBPB and CEBPD with Cell-Penetrating Peptides to Treat Brain and Other Cancers

**DOI:** 10.3390/cells12040581

**Published:** 2023-02-11

**Authors:** Lloyd A. Greene, Qing Zhou, Markus D. Siegelin, James M. Angelastro

**Affiliations:** 1Department of Pathology and Cell Biology, Columbia University Vagelos College of Physicians and Surgeons, New York, NY 10032, USA; 2Department of Molecular Biosciences, School of Veterinary Medicine, University of California Davis, Davis, CA 95616, USA

**Keywords:** brain cancer, glioblastoma, transcription factor, ATF5, CEBPB, CEBPD, dominant-negative, decoy, cell-penetrating, drug

## Abstract

Developing novel therapeutics often follows three steps: target identification, design of strategies to suppress target activity and drug development to implement the strategies. In this review, we recount the evidence identifying the basic leucine zipper transcription factors ATF5, CEBPB, and CEBPD as targets for brain and other malignancies. We describe strategies that exploit the structures of the three factors to create inhibitory dominant-negative (DN) mutant forms that selectively suppress growth and survival of cancer cells. We then discuss and compare four peptides (CP-DN-ATF5, Dpep, Bpep and ST101) in which DN sequences are joined with cell-penetrating domains to create drugs that pass through tissue barriers and into cells. The peptide drugs show both efficacy and safety in suppressing growth and in the survival of brain and other cancers in vivo, and ST101 is currently in clinical trials for solid tumors, including GBM. We further consider known mechanisms by which the peptides act and how these have been exploited in rationally designed combination therapies. We additionally discuss lacunae in our knowledge about the peptides that merit further research. Finally, we suggest both short- and long-term directions for creating new generations of drugs targeting ATF5, CEBPB, CEBPD, and other transcription factors for treating brain and other malignancies.

## 1. Introduction

The reader will well know the challenges of treating primary and recurrent malignancies of the brain and other organs and the need for new therapeutic approaches to these ends. Successful creation of new targeted cancer drugs often hinges on three key steps. First is the recognition of specific proteins that drive malignant properties such as tumor formation, growth, survival, metastasis and treatment resistance. Second is the development of strategies to interfere with the activity of such oncogenic proteins. Third is the design of suitable drug(s) to safely implement the strategies. In this review, we relate how ATF5 and then CEBPB and CEBPD were identified as targets for treatment of brain and other cancers. We then describe strategies that have been devised to interfere with their expression or activities and the consequences thereof on cancer cells. We highlight one such strategy, namely, the design of cell-penetrating dominant-negative decoy peptides that exploit the leucine zipper properties of ATF5 and/or CEBPB and CEBPD, and that selectively suppress the growth and survival of tumor cells. One such peptide is currently in clinical trials for brain and other cancers. While this review focuses on ATF5, CEBPB and CEBPD in the context of brain cancers, it is important to consider, as we briefly do here, the roles of these transcription factors in other types of cancers and how the targeted peptide drugs discussed here may be used to treat these as well.

## 2. A Brief Introduction to ATF5, CEBPB and CEBPD

ATF5, CEBPB (also referred to as C/EBPβ, C/EBPB or C/EBPbeta) and CEBPD (also referred to as C/EBPδ, C/EBPD or C/EBPdelta) are transcription factors that are members of the basic leucine zipper (bZIP) family that has been present and has diversified over the past billion years [1]. bZIP transcription factors are characterized by a basic DNA binding region followed by a leucine zipper (LZ) domain. The LZ possesses a leucine (or in some cases a valine) residue at every 7th position and forms coiled heptad repeats that confer the capacity for homo- or heterodimerization [2,3,4,5]. Such dimerization is obligate for DNA binding and transcriptional activity [2,3,4,5]. Interactions among family members are highly specific and are regulated by the sequences of their LZ domains [1,6,7]. As will be discussed here, the leucine zippers of ATF5, CEBPB and CEBPD are key features in the design of cell-penetrating decoy peptides to target them for brain and other cancers. The AlphaFold-predicted 3D structures [8,9] of human ATF5, CEBPB and CEBPD proteins, including their coiled bZIP domains, are shown in Figure 1.

The reader is directed to a number of excellent reviews that describe the general properties of ATF5 [10,11,12], CEBPB [13,14,15,16,17,18] and CEBPD [4,16,19].

## 3. ATF5: Recognition as a Target for Brain and Other Cancers

### 3.1. The Role of ATF5 in Growth and Differentiation of Neuroprogenitor Cells Suggests a Potential Role in Brain Cancers

Our interest in ATF5 and its potential role in brain cancers first arose from studies to identify genes regulated by nerve growth factor (NGF) in the context of neuronal differentiation. Angelastro et al. [20] employed Serial Analysis of Gene Expression (SAGE) as a quantitative unbiased approach (this was well before availability of RNAseq) to compare transcriptomes of rat PC12 pheochromocytoma cells before and after nine days of NGF treatment. Under such circumstances, PC12 cells transition from a proliferating neuronal-precursor-like state to a non-dividing phenotype with many properties of post-mitotic neurons, including extensive neurite outgrowth and electrical excitability [21]. Among the nearly 800 NGF-regulated transcripts detected, that encoding ATF5 was of particular interest based on its 25-fold drop in expression and identity as a transcription factor. This large decrease in expression was also observed at the protein level [22]. Subsequent studies to uncover ATF5′s role in neural differentiation revealed high levels of ATF5 transcripts and protein in mouse neural progenitor cells that give rise to neurons [22], astrocytes [23] and oligodendrocytes [24]. In contrast, there was little or no ATF5 in the corresponding differentiated cells. This paralleled the PC12 findings of high expression in dividing precursor cells and loss with differentiation. Constitutive ATF5 over-expression in PC12 as well as neuroprogenitor cells maintained them in a proliferating state [22] and blocked their transition to differentiated neurons, astrocytes [23] or oligodendrocytes [24]. These observations thus suggested that ATF5 maintains the undifferentiated, proliferating state of neural precursor cells and suppresses their differentiation. In line with this, ATF5^-/-^ mice exhibit behavioral abnormalities [25] and during embryonic development exhibit reduced numbers of cortical neuroprogenitor and radial glial cells [26].

Findings such as the fact that ATF5 suppresses the differentiation of neuroprogenitor cells and promotes their proliferation raised the question of its potential role in brain tumors. A particularly telling experiment that supported this idea was one in which retroviral infection was used to over-express ATF5 in dividing neuroprogenitor cells within the subventricular zone of newborn rats [23]. In contrast to cells infected with the control virus expressing only GFP, which differentiated and migrated into the cortex, those over-expressing ATF5-GFP continued to proliferate and, by 3 months, formed a large mass that protruded into the ventricles. These and other over-expression findings thus have supported the idea of a role for ATF5 in brain tumor malignancy.

### 3.2. ATF5 as an Anti-Apoptotic Factor

An additional set of early experiments uncovered another key activity of ATF5 that suggested a potential role in oncogenesis. Having noted that ATF5 (then referred to as ATFx) expression is greatly decreased in dying IL-3-dependent murine hematopoietic FL5.12 cells following cytokine deprivation [27], Persengiev et al. [28] explored its potential role in cell survival. It was found that ATF5 levels decreased in several other cell models of induced apoptotic death and that ATF5 over-expression blocked such death. It was also concluded that ATF5 over-expression did not affect cell proliferation, but rather it functions in an anti-apoptotic role. With respect to brain tumor cells, Sheng et al. [29] showed that ATF5 over-expression protected murine GL261 glioma cells from a variety of apoptotic stimuli including inhibitors of FGFR, EGFR, RAS or PI3K as well as from the kinase inhibitor sorafenib. These important findings thus indicate that ATF5 can drive survival of glioma cells as well as resistance to conditions or treatments that can otherwise promote apoptotic death.

### 3.3. Additional Evidence That Identifies ATF5 as a Potential Target for Treatment of Brain Cancers

Following the initial work that suggested a potential role for ATF5 in brain and other cancers, a variety of studies and approaches have provided support for the concept that ATF5 is a target for treatment of malignancies of the brain and other organs.

#### 3.3.1. ATF5 Expression in Brain Tumors

One potential criterion for identifying suitable targets for brain cancer therapeutics is that they are present and over-expressed in malignant cells. Multiple studies have addressed this issue in the case of ATF5.

1. *Detection by Immunostaining*. In a study by Angelastro et al. [30], sections from 29 resected GBM were immunostained for ATF5 expression. All 29 tumors surveyed showed positive nuclear ATF5 staining in the majority of cells. In contrast, no signal was detected in neurons either outside or within the tumors, and only a small number of stained non-tumor cells (thought to be reactive astrocytes) were seen. Consistent with this, six human and two rat GBM cell lines also showed positive nuclear staining, while early passage neonatal rat astrocytes had little or no signal. In another immunohistochemical study by Sheng et al. [29], sections from 38 GBM were assessed, of which 71% were reported positive for ATF5. There was no positive staining in the four normal brains that were surveyed. In additional work, Huang et al. [31] carried out ATF5 immunostaining on gliomas classified as either GBM, anaplastic or low-grade, as well as on normal cortex from glioma-free patients. It was reported that GBM and anaplastic tumors, but not low-grade tumors, showed significantly elevated ATF5 expression compared with normal brain tissue. Hu et al. [32] also compared ATF5 immunostaining in GBM, anaplastic and low-grade gliomas, and non-tumorous brain tissue. Positive nuclear and cytoplasmic staining was observed in 90% of glioma samples, 87% of anaplastic tumors, 29% of low-grade gliomas and <1% of non-tumorous samples. A study by Feldheim et al. [33] also evaluated ATF5 immunostaining in a set of samples from WHO grade II low-grade astrocytomas, as well as from normal brain and GBM. While normal brain showed no staining in astrocytes and weak predominantly cytoplasmic staining of neurons, tumor cells exhibited strong nuclear staining that was most pronounced in peri-necrotic palisades. Semiquantitative analysis of expression indicated highly significant differences in the tumors compared with normal brain and no significant difference in staining between GBM and low-grade astrocytomas. In addition to human and rodent tumor cells, ATF5 expression has also been examined in canine gliomas [34]. Immunostaining of two anaplastic oligodendrogliomas and two glioblastomas revealed significant staining in tumor cells and minimal staining in peritumoral normal brain.

2. *Detection by Western immunoblotting (WB)*. ATF5 protein expression in gliomas has also been evaluated by Western immunoblotting (WB). In addition to corroborating immunostaining, this approach has the advantage of assuring recognition of a single protein of the appropriate molecular size. Angelastro et al. [30] reported WB detection of ATF5 as a single 22 KD band in 8 glioma cell lines, but not in low passage astrocytes. Sheng et al. [29] also reported WB detection of ATF5 in GBM cell lines, as did Wang et al. [35] and Huang et al. [36]. WB quantification of ATF5 expression in tumor and normal brain samples by Huang et al. [31] revealed highly elevated levels in high-grade gliomas and no significant difference between normal brain and low-grade gliomas. The canine study cited above [34] also showed by WB significant elevation of ATF5 protein in gliomas compared with normal brain tissue.

3. *ATF5 mRNA levels in brain tumors*. *ATF5* mRNA levels have also been quantified in brain tumors by several groups. In evaluating such data, it must be kept in mind that protein expression does not necessarily reflect transcript levels. This is especially the case for ATF5, which, as with the more heavily studied ATF4, is subject to selective phospho-eIF2alpha-dependent translation under diverse conditions of cellular stress such as those which tumor cells are likely to encounter in situ [37,38]. Hua et al. [39] compared *ATF5* mRNA levels in normal brain tissue with those in low (I–II)- and high (III–IV)-grade gliomas and found a significant 3–4-fold increase in both low- and high-grade tumors. Huang et al. [31] reported no significant difference in *ATF5* mRNA levels between normal cortex and low-grade gliomas and a 5–6-fold elevation in anaplastic gliomas and GBM, while Wang et al. [40] observed significantly increased *ATF5* mRNA levels with increasing tumor grade with elevations of 50% in low-grade glioma, 2-fold in anaplastic glioma and 3-fold in GBM. Another study, by Feldheim et al. [33], quantified *ATF5* mRNA in a number of low-grade astrocytoma (WHO grade II) and GBM tumors as well as in normal brain. This revealed a 7-fold elevation in low-grade tumors and a 10-fold increase in GBM compared with normal brain, and with no significant difference between low-grade gliomas and GBM. Interestingly, no significant differences were detected between primary tumors and local or multifocal relapses or between tumors of different brain locations. In a further study, Hu et al. [41] analyzed open-source transcript data from The Cancer Genome Atlas (TCGA) to compare *ATF5* mRNA expression in large numbers of samples from normal brain, GBM, and low-grade gliomas. While there was significant over-expression in GBM, none was found for low-grade tumors.

4. *General conclusions.* Considered together, the above studies of ATF5 expression in gliomas indicate a consensus of significant ATF5 protein and mRNA elevation in GBM with little expression in normal brain. On the other hand, there appears to be a divergence in findings as to whether there is over-expression of ATF5 protein and mRNA in lower-grade gliomas. Resolving the latter issue for lower-grade gliomas is of importance, since it may have a bearing on whether targeting ATF5 and its binding partners would be a viable therapy for treatment of such tumors.

A related question that has not been well studied is the expression of ATF5 in brain tumors other than gliomas. Preliminary data from Lee [42] indicate immunostaining in medulloblastomas, but data on protein expression in other types of intrinsic brain malignancies are lacking. However, if one considers cancers that can metastasize to brain, these show moderate-to-strong nuclear staining at least outside the brain [43], and thus, when present in CNS tissue (or outside of it), these cancers are likely to be potential targets for the therapies described here.

#### 3.3.2. ATF5 Expression and Patient Survival

Given the over-expression of ATF5 in GBM and potentially lower-grade gliomas, several groups have queried whether ATF5 levels correlate with patient survival. Data from Dong et al. [44] first raised the possibility that *ATF5* mRNA levels in GBM might inversely correlate with mean survival time. Sheng et al. [29] compared survival times of 23 patients with GBM that were found to be either positive or negative for ATF5 immunostaining. Those with tumors having detectable ATF5 had significantly shorter survival times than patients with tumors in which ATF5 was not detected. The study by Feldheim et al. [33] compared survival of GBM patients with high or low levels of *ATF5* mRNA at the time of primary surgery (21 patients per group). In the first year after diagnosis, the group with lower expression had a significantly longer survival time and progression-free survival. However, these advantages diminished over longer times and were no longer significant by 18 months. It was noted that while the extent of tumor resection and temozolomide treatment was similar for both groups, the high ATF5 group was significantly older. Hu et al. [41] utilized data from the Chinese Glioma Genome Atlas (CGGA) to examine the relationship between *ATF5* mRNA levels and survival. For all glioma types taken together (163 GBM and 518 low-grade gliomas), there was a highly significant increase in mean survival time for those with low-expression ATF5 tumors. Considered together, these studies indicate a potential negative correlation between ATF5 expression and prognosis for GBM. For lower-grade gliomas, such a correlation is presently unclear.

#### 3.3.3. Evidence from Knockdown Studies Verifies ATF5 as a Potential Target in Brain Cancers

The studies cited above identify ATF5 as a potential target in GBM and possibly other brain cancers. A critical test of this is whether interference with ATF5 expression or function inhibits tumor cell growth, survival or other aspects of malignancy. Among the tools used to test this possibility has been knockdown with ATF5-targeted si- or shRNAs. Using si- and shRNAs that had been developed [22] to study ATF5′s role in neuroprogenitor cell growth and differentiation, Angelastro et al. [30] knocked down ATF5 in a rat (C6) and 4 human (U251, U138, U373 and T98G) GBM cell lines. This triggered significant levels of apoptotic death in each case. Importantly, ATF5 knockdown had no effect on survival of rat astrocytes that had been serially passaged and that expressed ATF5. Sheng et al. [29] reported that siRNA-mediated ATF5 knockdown induced apoptotic death of mouse GL261 GBM cells and of U87 GBM cells, as did Wang et al. [35,40]. In another study, ATF5 siRNA was also reported to reduce growth/survival of cultured U251 GBM cells [45].

To create an ATF5 siRNA that can be taken into cells by macropinocytosis and that can pass the blood–brain barrier, Huang et al. [36] encapsulated the siRNA-loaded calcium phosphate core with apolipoprotein E3-reconstituted high-density lipoprotein. Evidence was presented that the encapsulated siRNA is selectively taken up by Ras-expressing tumor cells. This reagent, designated ATF5-CaP-rHDL, significantly reduced the viability and caused apoptotic death of cultured rat C6 glioma cells and patient-derived glioblastoma-initiating cells, but not of cultured astrocytes. In both a rat C6 model and a patient-derived glioblastoma-initiating cell xenograft model in mice, ATF5-CaP-rHDL promoted apoptotic death of the implanted tumor cells and significantly extended animal survival. Analysis of behavior, animal weight, pathology and blood chemistry revealed no evident side effects of the treatment. Though not the subject of this review, such encapsulated siRNAs represent an intriguing alternative means for targeting ATF5 as well as CEBPB and CEBPD in GBM.

Taken together, multiple studies with si- and shRNAs indicate that reduction in ATF5 expression can be lethal to GBM cells both in vitro and in vivo, thus identifying the factor as a potential therapeutic target. Importantly, siRNA knockdown does not appear to affect viability of differentiated or activated astrocytes, thereby suggesting selectivity for malignant cells.

## 4. Dominant-Negative Constructs as a Strategy to Target ATF5 and Other Basic Leucine Zipper Proteins in Brain and Other Cancers

### 4.1. Development of Dominant-Negative ATF5 Constructs

Dominant-negative (DN) proteins are those that contain mutations such that their over-expression disrupts the function of their corresponding wild-type proteins or of other proteins that associate with them. This may occur, for example, for a transcription factor in which the mutated DN retains the capacity to dimerize with its appropriate binding partners, but in which the dimer, once formed, lacks the capacity to bind DNA and to regulate gene expression [46]. Dominant negatives can also promote the degradation of their wild-type binding partners [47,48].

There are multiple ways in which DNs can disrupt function of dimerization-dependent transcription factors such as ATF5, CEBPB and CEBPD (Figure 2). For example, in the case of factors that require homodimerization for activation, the corresponding DN mutants can directly block activity by forming inactive heterodimers with their corresponding wild-type targets (Figure 2A). For factors that do not homodimerize and that are activated by heterodimerization with a partner, DN mutants of either partner can directly block activity by forming inactive heterodimers with the other wild-type protein (Figure 2B). As will be discussed below, DN forms of ATF5 appear to act in this manner. A third example regards pairs of factors that are activated by forming both homo- and heterodimers. In this case, DN mutants of either factor block activity of both the homodimer pairs and the heterodimer pairs by forming inactive dimers with each of the wild-type factors (Figure 2C). As also discussed below, this appears to be the case for CEBPB and CEBPD and their corresponding DN mutant forms.

Vinson and colleagues [6] first suggested that DN forms of bZIP proteins might be designed to interfere with function by exploiting the dimerization properties of their leucine zippers. Further studies [49] indicated that although a modified leucine zipper alone had DN activity, and this could be substantially enhanced by adding an acidic amphipathic helix to the N-terminus of the zipper region. The N-terminal extension permitted the formation of a heterodimeric coiled coil with the basic DNA binding region of the targeted factor, thereby increasing interaction of the DN with the target factor and blocking the latter’s interaction with DNA [49,50]. The approach was verified with a DN form of CEBPA and was subsequently used, for example, to design a DN of the c-FOS protein that heterodimerizes with C-JUN [51].

Persengiev et al. [28] first reported use of a DN construct form of ATF5. Although the exact sequence was not given, it appeared to encode the C-terminal half of the protein including the basic DNA binding and leucine zipper domains. When transfected into HeLa and FL5.12 cells grown in the presence of survival factors, the construct promoted apoptosis, presumably by interfering with the survival-promoting activity of endogenous ATF5.

Angelastro et al. [22] also introduced a DN construct form of ATF5 in their study of ATF5′s role in neuronal differentiation. The sequence of the encoded protein (excluding an N-terminal FLAG tag) was given as **EQRAEELARENEELLEKEAEELEQENAELEGECQGL**EARNRELRERAESVEREIQYVKDLLIEVYKARSQRTRSA, where the DNA binding motif was replaced with an amphipathic acidic α-helical sequence (expressed in bold and containing leucine repeats at each seventh residue). The construct was predicted to interfere with ATF5 activity either by directly binding endogenous ATF5 and/or its heterodimerizing partners. The transfected construct accelerated NGF-promoted outgrowth by PC12 cells and phenocopied the capacity of ATF5 siRNA to promote differentiation of neuroprogenitor cells in vitro and in vivo [22,23,24].

### 4.2. DN-ATF5 Constructs and Effects on GBM Cells In Vitro and In Vivo

Findings of ATF5 over-expression in GBM as well as observations regarding the actions of ATF5 gain and loss of function on cell survival and neuroprogenitor proliferation and differentiation, led Angelastro et al. [30] to assess the effects of DN-ATF5 on a series of cultured GBM cell lines. All six human lines (U87, U251, DBTRG-05, U138, U373 and T98G) and one rodent line (C6) transfected with the DN construct showed high levels of apoptotic death. This included both p53+ and p53 mutated lines. In contrast, as with ATF5 siRNA, no such effects were seen on cultured early (ATF5 negative) or late (ATF5+ and proliferating) passage rat astrocytes. That death was apoptotic was indicated by observation of nuclei with apoptotic morphology and inhibition of death by the pan-caspase inhibitor BAF. These findings were thus consistent with the study discussed above by Persengiev [28] with HeLa and FL5.12 cells as well as with the above reviewed ATF5-siRNA data.

To evaluate the effects of the DN-ATF5 construct on tumor and non-tumor cells in vivo, Angelastro et al. [30] implanted C6 glioma cells into adult rat striatum and then, when the tumors formed, injected them with retroviruses expressing either GFP-DN-ATF5 or GFP alone. The retroviruses infected proliferating cells both within and outside of the tumors. Cells within the tumors infected with DN-ATF5 showed high levels of cell death as indicated by TUNEL staining. In contrast, only very low levels of TUNEL staining were seen in tumor or non-tumor cells infected with GFP alone or in non-tumor cells infected with DN-ATF5. Thus, DN-ATF5 showed selective killing of GBM cells in a homotopic tumor model without evident effects on nearby proliferating non-tumor cells.

As an alternative approach to assessing the actions of DN-ATF5 on glioma cells in vivo, Arias et al. [52] established bi-transgenic “TET-OFF” mice in which expression of DN-ATF5 in cells expressing glial fibrillary protein (GFAP) is inducible by withdrawing doxycycline from the diet. Since GFAP is highly expressed in neuroprogenitor and neural stem cells that can give rise to gliomas as well as in gliomas themselves, such mice provided the opportunity to monitor effects of DN-ATF5 on both glioma formation and persistence. Significantly, DN-ATF5 induction in embryonic or adult GFAP+ cells showed no evident cell death or other effects. To induce experimental malignant gliomas, juvenile mouse SVZ and corpus callosum were injected with retrovirus expressing PDGF-B and p53 shRNA [53,54]. While 15/16 control animals in which DN-ATF5 was turned off at the time of tumor induction developed tumors that expressed GFAP and ATF5, only 1/7 animals in which DN-ATF5 was on at the time of tumor induction developed a glioma. In additional studies, to assess the effect of DN-ATF5 on established tumors, the DN trans gene was induced only after the onset of tumor formation. All control mice in which DN-ATF5 was maintained in an off state developed gliomas by 150 days after tumor initiation, with 40% showing moribund behavior by the 6-month experimental endpoint. In contrast none of those in which DN-ATF5 was induced exhibited detectable tumors and all survived.

## 5. Drugging ATF5 Signaling in Brain and Other Tumors with Cell-Penetrating DN-ATF5 (CP-DN-ATF5): Design, Efficacy and Safety of CP-DN-ATF5 as a Potential Drug

Thus far, we have reviewed ATF5 in the context of the first two key elements of cancer drug discovery, namely, its identification as a potential target, and secondly, development of strategies to interfere with its function. In what follows, we recount the third step in which this information has been exploited to design potential therapeutics for brain and other cancers.

A major challenge for the use of peptides such as DN-ATF5 is to promote their entry into living cells as well as to endow them with the capacity to pass through tissue barriers such as the blood-brain barrier. One means to achieve this is to incorporate a suitable cell-penetrating domain into the peptide [55]. For DN-ATF5, an N-terminal 16 amino acid “penetratin” domain [56] was included in the peptide design as described by Cates et al. [57]. As a step to reduce potential aggregation of the peptide [58] and to shorten it for ease of synthesis, the last 25 amino acids of the parent ATF5 protein were omitted [57]. This characteristically deleted portion of ATF5 includes two valine/valine C-terminal heptad repeats within the leucine zipper domain (see Figure 3). Active forms of CP-DN-ATF5) were generated by either bacterial expression [57] or by commercial synthesis [59].

Confocal microscopy of tagged CP-DN-ATF5 revealed its rapid uptake by cultured tumor cells where it was detected in both the nucleus and cytoplasm [57]. Moreover, when delivered intraperitoneally to mice (four injections of 1 mg/kg at 1–2 h intervals) with brain gliomas induced as described above [52], extensive uptake was evident in both brain and tumor cells when assessed at 16 h after delivery, and was still detectable at reduced levels at 64 h [57]. In terms of efficacy, the study of Cates et al. [57] revealed that CP-DN-ATF5 caused apoptotic death of cultured C6 and U87 glioma cells and of three lines of human glioma-initiating cells. It also triggered TUNEL staining in induced glioma cells in vivo within 24 h of treatment. Importantly, although the peptide was also taken in by normal brain cells, little or no TUNEL staining was observed in such non-tumor tissues.

Additional in vivo experiments used a treatment protocol as above, but with two subcutaneous treatments 5 days apart [57]. Mice bearing induced gliomas and assessed by MRI showed tumor regression or loss after 8 days of CP-DN-ATF5 treatment, and no recurrence was detected 6–13 months afterwards. The absence of tumors in the treated animals was also verified by histology. No such regression or loss of tumors was seen in vehicle controls. In terms of survival, all treated mice survived to the 180 day endpoint of the study, whereas 67% of control mice died within this time. An intracranial xenograft model using luciferase-expressing U87 GBM cells also showed a highly significant reduction in tumor volume in peptide-treated animals.

An important feature of the above in vivo studies with CP-DN-ATF5 was that no significant toxicity to normal tissue was observed. Full body necropsies carried out 2 days or 6 months after treatment revealed no evident pathology and a liver-kidney serum chemistry panel carried out 1 day after treatment showed no damage to either organ [57].

A study by Karpel-Massler et al. [59] further characterized the actions of CP-DN-ATF5 on gliomas and other types of cancer cells. In the realm of brain tumors, in addition to U87 cells, the peptide promoted apoptotic death of cultured GBM lines T98G, U251, LN229, SF188 (pediatric GBM), and GBM12 (PDX) as well as of glioma lines MGPP3 (murine proneural) and NCH644 (glioma stem-like). In contrast, a mutated form of CP-DN-ATF5 in which key leucine residues were replaced with glycines, showed greatly diminished activity on T98G cells. Several GBM xenograft models were also investigated including U87 (flank), U251 (flank), and GBM12 (intracranial) cells. Compared with control penetratin peptide alone, intraperitoneally delivered CP-DN-ATF5 (50–150 mg/kg 1–4x/week, depending on model), significantly reduced tumor growth in all models. In the case of the intracranial GBM12 model, animal survival was determined and was significantly prolonged. Histological comparison of tumor-free tissues in peptide and control-treated animals showed no alterations in brain, lung, kidney, heart, liver, spleen, and intestine. There were also no effects on body weight.

## 6. DN–ATF5 Targets bZIP Transfection Factors CEBPB and CEBPD, but Not ATF5

As discussed above, DN proteins can act by various means including interfering with activity of the parental protein and/or that of its heterodimerizing partners. On the basis of experiments on the interaction of in vitro-translated ATF5 (then referred to as ATF-7) with radiolabeled CRE oligonucleotide probe, Peters et al. [60] reported that ATF5 binds DNA as a homodimer. Ciaccio et al. [61] confirmed that the purified ATF5 bZIP domain associates with a CRE DNA oligonucleotide and suggested that this requires disulfide bond formation. On the other hand, they observed that purified ATF5 exists mainly as a monomer in solution.

On the basis of such findings, it was anticipated that ATF5 itself might be one of the potential targets for DN-ATF5. A study by Sun et al. [62] tested this idea as well as what other proteins the peptide might target in living cells. First, it was observed that neither siRNA-mediated ATF5 depletion nor engineered disruption of the ATF5 gene affected the survival of HAP1 chronic myelogenous leukemia cell lines. Moreover, transfection with DN-ATF5 or exposure to CP-DN-ATF5 promoted similar levels of apoptotic death in WT and ATF5^-^ HAP1 cells regardless of ATF5 expression. This suggested that DN-ATF5 must have relevant targets other than, or in addition to, ATF5 itself. To identify cellular targets of DN-ATF5, PC3 prostate cancer cells were transfected with GFP-FLAG-DN-ATF5, and extracts were subjected to pull-down with anti-FLAG beads followed by SDS-PAGE and then LC-MS/MS. Control-based filtering from two independent experiments returned three robust “hits”. These were the basic leucine zipper transcription factors CEBPB and CEBPD and the coiled-coil domain protein CCDC6. Significantly, there was no signal for ATF5 protein itself outside of the leucine zipper present in the DN-ATF5 used for the pulldown. Pulldown-Western blot studies confirmed that transfected DN-ATF5 associates with CEBPB, CEBPD and CCDC6 in T98 GBM cells, with CEBPB and CEBPD in LN229 GBM cells and with CEBPB in U87 GBM cells (in which CEBPD protein was not detected). CEBPB requires phosphorylation at Thr235 to bind DNA [63], and this species too was pulled down with DN-ATF5 in all 3 GBM lines. In contrast, a DN-ATF5 mutant (in which leucine zipper leucines were replaced with glycine), failed to pull down CEBPB, pCEBPB or CEBPD in the same GBM cells. Interestingly, another mutated form of DN-ATF5 in which the extended leucine zipper was deleted also associated with CEBPB, pCEBPB and CEBPD in pulldown assays with T98 and LN229 cells, thus indicating that the extended leucine zipper formed by mutating the DNA binding domain is dispensable for interaction with these proteins. Additional experiments with T98G cells showed that synthetic CP-DN-ATF5 as well as a synthetic form of the peptide lacking the extended leucine zipper successfully competed with transfected tagged DN-ATF5 for association with CEBPB, CEBPD and CCDC6.

Association of DN-ATF5 with CEBPB and CEBPD raised the important question of whether it also affects their transcriptional activities. Experiments by Sun et al. [62] with T98G cells showed that CP-DN-ATF5 significantly reduced nuclear levels of active CEBPB. In addition, CP-DN-ATF5 suppressed expression of the CEBPB and CEBPD targets *IL6* and *IL8* in T98G and LN229 GBM cells as well as in WT and ATF5^-^ HAP1 cells, and significantly reduced CEBPB and CEBPD occupancy of the *IL6* and *IL8* promoters in T98G cells. Finally, CP-DN-ATF5 significantly reduced the activity of a luciferase reporter driven by the *IL6* promoter, which contains a CEBP consensus binding site for CEBPB and CEBPD. These findings thus indicate that CP-DN-ATF5 not only associates with CEBPB and CEBPD, but also interferes with their transcriptional activities in GBM cells.

Given that DN-ATF5 triggers apoptosis of glioma and other cancer cell types, identification of CEBPB, CEBPD and CCDC6 as direct DN-ATF5 targets suggested that these too may play required roles in tumor cell growth and survival. To test this, the three targets were individually knocked down in T98G cells with siRNAs [62]. Knockdown of CEBPB and CEBPD, but not of CCDC6, significantly increased apoptotic cell death. A similar elevation of apoptotic death was achieved by CEBPB and CEBPD knockdown in LN229 and GBM22 cells. By contrast, CEBPB or CEBPD knockdown in cultured normal astrocytes, despite interfering with *IL6* expression did not affect their survival.

Taken together, these findings identify CEBPB and CEBPD, but not ATF5 itself, as direct targets of DN-ATF5 and indicate that the selective apoptotic activity of DN-ATF5 on glioma and other cancer cells is mediated by direct interference with CEBPB and CEBPD function and, as is likely, by indirect suppression of ATF5 activity by depriving it of its hetero-dimerization partners CEBPB and CEBPD.

## 7. CEBPB and CEBPD as Additional Targets for Treatment of Gliomas and Other Cancers

The discovery that DN-ATF5 is active on brain and other cancers due to its interference with the activities of CEBPB and CEBPD is consistent with a number of reports in the literature that identify these transcription factors as potential targets for treatment of gliomas. These are reviewed below and strongly support CEBPB and CEBPD as brain cancer targets.

### 7.1. CEBPB/CEBPD Expression and Patient Prognosis

Drawing either on primary data or data from public data bases, multiple studies have reported that CEBPB protein and/or mRNA are elevated in gliomas and that high expression correlates with poor patient outcome [64,65,66,67,68,69]. Similar findings have been put forward for CEBPD [66,70,71]. Regarding the relationship between tumor grade and expression, data indicate that high-grade (GBM) tumors express higher levels of CEBPB [64] and CEBPD [70,71]. Such conclusions are consistent with the poorer prognosis of GBM patients compared with those with lower-grade tumors. Among GBM, there also appears to be a correlation of CEBPB expression with subtype and outcome. In particular, higher expression of CEBPB protein or mRNA levels are found in GBM with the highly aggressive mesenchymal phenotype [65,67,68,72]. Cooper et al. [66] also distinguished differences in CEBPB and CEBPD expression within GBM tumors, with highly upregulated expression of the two proteins in hypoxic, perinecrotic “pseuopalisading” cells. It was suggested that this might contribute to the poor prognosis associated with these factors.

### 7.2. CEBPB and CEBPD and Transition to the Mesenchymal Phenotype in GBM

The aggressive nature and treatment resistance of GBM with a mesenchymal signature and the transition of less aggressive tumor cells to this phenotype have led to studies of the underlying regulators of this state. Carro et al. [65] employed computational methods to identify “master regulators” of the mesenchymal phenotype and identified CEBPB and CEBPD, along with STAT3, at the top of a pro-mesenchymal regulatory hierarchy. Follow-up studies focused on CEBPB, though it was noted that CEBPB and CEBPD form stoichiometric homo- and heterodimers with identical DNA binding specificity and redundant transcriptional activity. Overexpression of CEBPB along with STAT3 in neural stem cells caused loss of neuronal differentiation, manifestation of a fibroblast-like morphology, induction of mesenchymal genes and enhanced migration in a wound assay. Conversely, siRNA- and shRNA-mediated CEBPB knockdown in SNB19 GBM cells showed suppression of the mesenchymal signature. Importantly, SNB19 cells in which CEBPB was silenced with shRNA were significantly less effective in forming intracranial tumors in xenograft experiments. Similar results were achieved with GBM-patient-derived brain tumor-initiating cells. Histological analysis revealed that CEBPB expression was significantly correlated with mesenchymal markers in a series of glioma specimens and with poor patient outcome.

Several additional studies link CEBPB to transition of GBM to the mesenchymal phenotype and examined how this is regulated. Halliday et al. [72] studied the response of PDGF-B-induced mouse gliomas to radiation. It was observed that the tumor cells underwent a shift from a proneural to mesenchymal phenotype within a few hours of irradiation in an apparent p53-independent manner. Such a shift has been associated with GBM radioresistance. Transcriptome analysis revealed that the mesenchymal shift correlated with highly elevated expression of CEBPB as well as of STAT3. Minata et al. [73] also described a role for CEBPB in the response of GBM to radiation and the relation thereof to tumor recurrence. It was found that human GBM contain individual populations of cells with either a proneural or mesenchymal signature and that the former tend to be at the tumor margins. Moreover, radiation, which stimulates CEBPB expression, induced conversion of proneural cells to the more aggressive mesenchymal phenotype with enhanced stem cell and tumor-initiation properties, as well as radiation resistance. Significantly, siRNA-mediated knockdown showed that CEBPB was required for this conversion. In this context, CEBPB transcriptionally elevated expression of the CD109 protein, which in turn activates oncogenic signaling through the YAP/TAZ pathway. In additional work, Yin et al. [74] studied transglutaminase 2 (TGM2) which was upregulated in perinecrotic areas of GBM and promotes the mesenchymal transdifferentiation of patient-derived GBM stem cell lines and their growth and malignancy in orthotopic xenograft models. A pathway was described in which TGM2 crosslinks and drives degradation of DDIT3 (CHOP/GAD153), an inhibitor of CEBPB activity and expression, resulting in CEBPB upregulation and promotion of the mesenchymal phenotype. Conversely, TGM2 inhibition in GBM stem cell lines led to CEBPB downregulation, loss of mesenchymal properties, cell death and suppression of growth in orthotopic xenografts. Taken together, these studies thus identify CEBPB and CEBPD as critical players in the induction of the more aggressive and treatment-resistant mesenchymal GBM phenotype.

### 7.3. CEBPB/CEBPD and Glioma Cell Survival, Growth and Invasive Behavior

A key issue pertinent to targeting CEBPB and CEBPD is whether interference with their activities or expression affects relevant glioma cell properties such as survival, growth and invasive behavior. In this context, Homma et al. [64] knocked down CEBPB in U251 glioma cells with siRNA and found that this reduced cell numbers, cell invasion through Matrigel, and secretion of CEBPB target and potential contributor to GBM growth, IL8. Aguilar-Morante et al. [75] used shRNA to silence CEBPB in murine GL261 and human LN18 GBM cells. This significantly reduced both cell survival and viability and inhibited motility in a scratch assay. Moreover, GL261 cells in which CEBPB was knocked down formed significantly smaller tumors than control cells when implanted into mouse brains. Mice with such tumors also had a significantly longer survival time than those with control tumors. In a follow-up study, Aguilar-Morante et al. [76] reported that CEBPB knockdown also reduced growth of T98G cells and of stem cells derived from GL261 cells that form self-renewing neurospheres. Evidence was also presented that CEBPB promotes GBM cell motility and invasion in culture by inducing S100A4 expression. Di Pascale et al. [77] demonstrated the role of CEBPB in transcriptional activation of miR-138, which they characterized as a microRNA that promotes survival and proliferation of glioma stem cells. Consistent with this, U87MG cells in which CEBPB was knocked down showed diminished miR-138 expression as well as decreased growth in an orthotopic xenograft model, and concomitant decrease in cell proliferation and elevation of apoptotic markers. Regarding CEBPD, Wang et al. [70] investigated the role of CEBPD in regulation of glioma vasculogenic mimicry (VM), a mechanism of tumor microcirculation that does not depend on endothelial cells. A pathway was uncovered in which SUMO-conjugating enzyme UBE21 promotes the SUMOylation of RNA-binding protein PUM2, destabilizing it, thereby stabilizing CEBPD, which in turn induces desmosome component DSG2, which promotes vasculogenic mimicry. It was also reported that CEBPD knockdown significantly inhibited the capacities for migration, invasion, and VM in cultures of U251 and U273 glioma cells. CEBPD knockdown in the two lines also decreased their growth in subcutaneous xenograft models and prolonged animal survival. Another study [78] found that spheroids formed from U87MG cells showed elevated cell death and smaller size when knocked down for CEBPD. Lin et al. [79] reported that CEBPD knockdown reduced cell viability and promoted apoptosis in cultures of T98G and U373MG cells. T98G cells with stable knockdown of CEBPD showed significantly inhibited tumor growth in flank xenografts and elevated TUNEL staining. CEBPD knockdown in T98G and U373 cells also reduced the oxygen consumption rate and caused an accumulation of H_2_O_2_. Evidence was given that CEBPD directly regulates the expression of catalase (CAT) and that CEBPD regulates cell survival via catalase induction, presumably by promoting H_2_O_2_ breakdown.

### 7.4. CEBPB/CEBPD and Glioma Stem Cell Renewal and Growth

Self-renewing glioma stem cells/glioma-initiating cells are postulated to participate in glioma genesis, recurrence and treatment resistance [80]. Several studies have indicated roles for CEBPB/CEBPD in the survival and growth of such cells. As noted above, CEBPB knockdown reduced formation of self-renewing neurospheres derived from GBM stem cells [76] and reduced formation of intracranial tumors by GBM-patient derived brain tumor-initiating cells [65]. In other work, Wang et al. [78] studied IL1B-stimulated glioma spheroid formation by T98G and U373 GBM cells and by glial stem cells from patient tumors. In each case, CEBPD knockdown significantly suppressed spheroid formation and growth. Data was presented indicating that CEBPD promoted spheroid formation by inducing PDGF-A. Additionally, Wang et al. [71] reported that CEBPD directly upregulates stem-cell related factors OCT4, SOX2, and NANOG in U87MG cells while CEBPD knockdown enhanced cell death and suppressed growth of the cells in spheroids.

### 7.5. CEBPB/CEBPD and Temozolomide Resistance in Gliomas

In addition to the work cited above regarding the roles of CEBPB in resistance to radiation, several studies examined the roles of CEBPB and CEBPD in resistance to the GBM standard of care drug temozolomide. (TMZ). Gao et al. [81] studied the effects of hypoxia on U87 cells and reported that this elevated CEBPB mRNA and protein levels as well as conferring resistance to TMZ. This resistance was overcome by siRNA-mediated CEBPB knockdown. In additional work, Gao et al. [82] identified an lncRNA, PDIA3P1, that was responsible for resistance to TMZ in multiple GBM cell lines and that correlated with the mesenchymal phenotype. It was found that PDIA3P1 associates with CEBPB and stabilizes it by preventing its ubiquitination by the MDM2 E3 ubiquitin ligase. Significantly, evidence was presented that PDIA3P1-dependent CEBPB upregulation was required for TMZ resistance. In a further study, Wang et al. [71] presented findings that the ABC efflux transporter ABCA1 is a direct CEBPD target and that its CEBPD-dependent elevation participates in TMZ resistance.

### 7.6. Regulation of CEBPB/CEBPD in Gliomas by Oncogenic Drivers

In addition to understanding the roles of CEBPB/CEBPD in gliomas, several studies have explored their relationship to oncogenic drivers of such tumors. Selagea et al. [83] showed that CEBPB in U87 cells is upregulated and activated by EGFR, a receptor protein that is often over-expressed and/or mutated in GBM and that contributes to GBM malignancy. Significantly, CEBPB, in turn, upregulated EGFR, suggesting a malignant loop. Lei et al. [69] reported that CEBPB was upregulated in U87MG cells by overexpression of EGFR or oncogenic mutated EGFRvIII, and by the presence of reactive oxygen species (ROS).

## 8. ATF5, CEBPB and CEBPD as Targets for Treatment of Non-Brain Cancers

Taken together, the findings reviewed here thus far identify CEBPB and CEBPD as well as ATF5 as attractive proteins for targeted therapy of brain tumors. While not the direct subjects of this review, it is highly relevant to note that these factors have also been implicated as potential targets in a wide variety of other tumor types. This includes colorectal [84,85,86,87,88,89], non-small cell lung [90,91,92], breast [45,90,93,94,95,96,97,98,99,100,101,102,103,104,105], ovarian [106,107,108,109], bladder [110,111,112], gastric [113,114], prostate [115,116,117,118,119], gallbladder [120], pancreatic [121,122], skin [123,124] and cervical [125] cancers; nasopharyngeal [126,127,128], head and neck [129], oral [130], and esophageal [131,132,133,134] squamous cell carcinomas; cutaneous [135] and uveal [136] melanoma; alveolar rhabdomyosarcoma [137]; lung adenocarcinoma [138,139]; neuroblastoma [139,140]; Ewing sarcoma [141]; urothelial carcinoma [142,143,144]; acute myeloid [145,146,147], acute lymphoblastic [148,149,150], acute promyelocytic [151] and chronic lymphocytic [152] leukemias; multiple myeloma [153]; ALK-positive anaplastic large cell [154] and follicular [155] lymphomas.

## 9. Targeting ATF5, CEBPB and CEBPD with NEXT Generation Cell-Penetrating Leucine Zipper peptides: Bpep and Dpep

### 9.1. Potential Advantages of Directly Targeting ATF5, CEBPB and CEBPD Simultaneously

As we have reviewed thus far, there is abundant evidence to support targeting of ATF5, CEBPB and CEBPD individually for therapeutic treatment of gliomas and an array of other cancer types. What is not presently clear is whether individual tumor cells are equally susceptible to inhibiting these transcription factors one at a time. It follows that an even more effective strategy may be to simultaneously suppress activities of all three. In essence, this is apparently the case for CP-DN-ATF5. As recounted here, CP-DN-ATF5 appears to directly associate with and block homo- and -heterodimeric activities of both CEBPB and CEBPD and to indirectly interfere with ATF5 activity by depriving it of its heterodimerization partners CEBPB and CEBPD. However, as noted above, CP-DN-ATF5 does not appear to directly bind ATF5 itself.

Why, then, might there be value in directly and simultaneously targeting ATF5 along with CEBPB and CEBPD? As reviewed above, multiple studies indicate that directly depleting ATF5 in cancer cells suppresses their growth and survival. Not all of these effects, however, may be due to interference with ATF5′s traditional roles in gene transcription. For instance, ATF5 is reported to bind GABA_B_ receptors [156], which have been implicated in several malignancies [157,158], and DISC1 (disrupted in schizophrenia 1 [159,160], which has been connected to promoting multiple malignant properties of GBM cells [161]. ATF5 is additionally reported to play an essential structural role in interaction of centrioles with pericentriolar material by associating with polyglutamylated tubulin and pericentrin [162]. ATF5 knockdown with shRNA led to centriole fragmentation and genomic instability [162]. In a different role, ATF5 has been described as an evolutionarily conserved mediator of the mitochondrial unfolded protein stress response [163] that is postulated to play a protective role in glioma and other cancer cells [11,164,165]. Although ATF5′s role in the mitochondrial unfolded protein response appears to be transcriptional, its binding partners in this activity have yet to be defined. Additionally, as discussed below, ATF5 as well as CEBPD and CEBPB may each associate with additional transcription factors that also play important global or context-dependent roles in malignancy.

### 9.2. Dpep and Bpep as New Cell-Penetrating Peptides to Simultaneously Target ATF5, CEBPB and CEBPD and as Potential Therapeutic Treatments for Brain and Other Cancers

In the context of developing agents that simultaneously directly inhibit ATF5, CEBPB and CEBPD, we reasoned that if DN-ATF5 binds CEBPB and CEBPD, then DN forms of CEBPB and CEBPD should reciprocally form inactivating heterodimers with ATF5 as well as form inactivating homo- and heterodimers with CEBPB and CEBPD (see Table 1). Consistent with this idea, association of ATF5 with at least CEBPB has been reported by Zhao et al. [166] using yeast two hybrid technology and pulldown assays. We also reasoned that such DN peptides should be at least as effective as DN-ATF5 in suppressing the survival and growth of cancer cells. As shown in Figure 4, computational docking models [167] for interaction of a peptide corresponding to the first 21 amino acids of the CEBPD leucine zipper support its association with the leucine zippers of ATF5, CEBPB and CEBPD.

To test these hypotheses, Zhou et al. [168] prepared constructs expressing DN decoy forms of CEBPB and CEBPD. Given the observation that a truncated form of DN-ATF5 in which the extended leucine zipper was deleted retained the capacity to associate with CEBPB and CEBPD in cells [62], the DN decoy forms consisted only of the respective CEBPB and CEBPD leucine zipper domains [168]. Both constructs, but not mutant constructs in which the key leucine residues were replaced by glycines, triggered apoptosis of T98G GBM and HCT116 colon cancer cells. This was followed [168] by design and synthesis of cell-penetrating peptides in which a penetratin sequence was placed N-terminal to the CEBPB and CEBPD leucine zippers (see Figure 3). The peptides, designated Dpep and Bpep, proved to cause apoptosis and disrupt growth of 3 human GBM lines (T98G, LN229, U251) and one mouse GBM line (MGPP3), as well as lines derived from human malignant melanoma, breast, lung, colon and myelogenous leukemia cancers. Peptides with leucine zipper mutations exhibited greatly decreased potency, highlighting the importance of the zipper domain for activity. Both Dpep and Bpep exhibited EC_50_ values averaging about 20 µM in monolayer cultures (which is 5–10-fold lower than that observed with CP-DN-ATF5) and of about 500 nM in colony-forming assays. In contrast, no effect was observed on growth or survival of three types of non-transformed cells, including astrocytes. Dpep and Bpep worked additively when combined, suggesting that the two act via similar mechanisms as might be anticipated. In consonance with past reports cited above in which interference with ATF5, CEBPB or CEBPD expression suppressed migration of cancer cells. Dpep and Bpep also inhibited migration of T98G and breast cancer MDA-MB-231 cells in scratch assays. Moreover, consistent with interference with CEBPB, CEPBD and ATF5 activities, in T98G and other non-glioma cancer lines, Dpep and Bpep suppressed expression of direct CEBPB/CEBPD targets *IL6* and *IL8* and of direct ATF5 and CEBPB target asparagine synthetase (*ASNS*).

Though not tested on gliomas, intraperitoneally delivered Bpep and Dpep (20–50 mg/kg, 3x/week) also showed anti-tumor activity when tested in subcutaneous xenograft models of melanoma and colon cancer [168]. This was manifested by rapid onset of TUNEL-positive staining in the tumors (but not surrounding tissues or in tumors of vehicle treated animals), significantly decreased tumor growth, and significant prolongation of animal survival. Observations of animal behavior, weight and histology of multiple organs revealed no evident side effects. Thus, like CP-dn-ATF5, Bpep and Dpep show both efficacy and apparent safety in animal cancer models. Considering their activity on GBM cells in vitro, it seems likely they will be effective on them in vivo, but this remains to be established.

## 10. ST101

ST101 is a cell-penetrating leucine zipper peptide developed by Sapience Therapeutics that appears to have promise for clinical treatment of brain and other cancers [169]. In 2016 Sapience licensed the CP-DN-ATF5 technology from Columbia University and then designed and patented a peptide, ST101, that Sapience has designated as a “CEBPB antagonist”. Though prior publications on CP-DN-ATF5, Bpep and Dpep were not cited, as described by Darvishi et al. [169] ST101 appears to have both significant similarities and differences compared with these peptides. In contrast to CP-DN-ATF5, Dpep and Bpep, ST101 is composed of D-amino acids. Among other potential advantages, the use of D-amino acids provides protection from proteolytic degradation as shown by resistance to pepsin or trypsin treatment in vitro. The published sequence of ST101 [169] is shown in Figure 3. Expressed as the mirror image for comparison with L-amino acid sequences, the ST101 sequence bears an evident strong resemblance to CP-DN-ATF5, Dpep and Bpep in the penetratin domain, while the leucine zipper sequence shows similarity to the leucine zipper portion of CP-DN-ATF5, including preservation of crucial leucine residues and truncation after the first valine residue in the ATF5 leucine zipper (Figure 3B). As might be anticipated by the resemblance of its truncated leucine zipper sequence to those in ATF5 and CP-DN-ATF5, ST101 was reported to bind the CEBPB leucine zipper in solution [169]. In contrast, there was a weak or undetectable association with the CEBPG or ATF5 leucine zippers. ST101 was also able to displace the ATF5 association with plate-bound CEBPB in an ELISA assay. Taken together, the leucine zipper sequence of ST101 and its association with CEBPB and not ATF5 suggest that it acts more like DN-ATF5 than DN-CEBPB or DN-CEBPD. Possible interaction of ST101 with CEBPD was not reported, but it seems likely (Table 1), given the reported association of CEBPD with DN-ATF5 [62].

Among the actions reported for ST101 is stimulation of ubiquitin-dependent proteasomal degradation of CEBPB [169]. At a concentration of 20 µM ST101 applied for 24 h, CEBPB levels dropped by about 30–40% in U251 GBM cells as well as in a colon cancer line. This effect could contribute in part to the actions of the peptide on cancer cell survival and growth described below. In contrast, Sun [62] reported no effect of CP-DN-ATF5 on CEBPB levels in 3 GBM lines after 24 h of treatment, while Karpel-Massler [59] found that CP-DN-ATF5 significantly depletes ATF5 protein in GBM lines at 48 but not 24 h of exposure, and it does so by decreasing its stability.

Like CP-DN-ATF5, owing to its penetratin-like sequence, ST101 is rapidly taken up by cultured tumor cells and when delivered peripherally into mice, can pass the blood–brain barrier and undergo cellular uptake [169]. Presumably due to its capacity to interfere with the activity of CEBPB (and perhaps additional binding partners such as CEBPD), ST101 compromises the growth and survival of brain and other tumor cells in vitro and in vivo. T98G, U87 and U251 GBM cells as well as a variety of other tumor cell types showed a loss of viability with an average EC_50_ of approximately 2 µM ST101, as evaluated by high-content imaging of Annexin V/PI staining [169]. By comparison, EC_50_ values for CP-dn-ATF5 on GBM lines range from about 100 to 200 µM. This difference could be due in part to the use of D-amino acids in ST101 and/or to the presence of the extended leucine zipper in CP-DN-ATF5. As noted above, the average EC_50_ values of Dpep and Bpep (which lack an extended leucine zipper) as measured by cell counting in monolayer cultures are in the range of 20 µM and approximately 500 nM in colony-forming assays [168].

ST101 showed impressive anti-tumor activity in subcutaneous xenograft models [169]. Given subcutaneously 3x/week at 50 mg/kg for 3 weeks, ST101 strongly suppressed the growth of xenografted U251 G×BM cells in a subcutaneous model. It was reported that by gross evaluation, 3/6 animals were tumor-free at the end of the 90-day study, while the remaining tumors continued to increase in size. Similar treatment of melanoma, breast, prostate and lung tumors with 25 mg/kg ST101 also inhibited growth. However, with the exception of A549 lung tumor cells (evaluated at 70 days), the tumors in treated animals continued to increase in size over time. No survival data were presented for any of the models. It was reported that there was no significant effect of treatment on animal body weight. Importantly, mice treated intravenously with 10 mg/kg of ST101 once per week for 6 weeks showed no detectable titer to the peptide as assessed by ELISA.

Additional findings regarding ST101 as presented in poster form can be found on the Sapience Therapeutics website.

## 11. Mechanisms of Action of CP-DN-ATF5, Bpep, Dpep and ST101 on Brain and Other Tumor Cells

### 11.1. The Peptides Promote Tumor Cell Apoptosis

It is clear from the literature reviewed above that a major response of brain and other tumor cells to ATF5, CEBPB and CEBPD knockdown or interference with activity is the appearance of apoptotic cell death. As anticipated, this is also the case for CP-DN-ATF5, Bpep, Dpep and ST101 documented in vitro by multiple means including examination of nuclear morphology, inhibition by caspase inhibitors, caspase activation/cleavage, PI/Annexin V flow cytometry, flow analysis of sub-G1 DNA, loss of mitochondrial membrane potential, and high-content screening for Annexin V/PI staining [59,62,168,169]. Promotion of apoptotic cell death also appears to occur in vivo in which TUNEL staining rapidly appeared in cells of an induced orthotopic mouse glioma model after treatment with CP-DN-ATF5 [57] and in a subcutaneous melanoma xenograft after Dpep treatment [168].

### 11.2. Dysregulation of Cell Pro- and Anti-Apoptotic Proteins BCL2, MCL1, Survivin and BMF

Among the described events by which the peptides trigger apoptosis is via dysregulation of genes and proteins that regulate tumor cell survival. Survival protein BCL2 has been described as a target of ATF5 and CEBPB [152,170], while survival protein MCL1 is reported to be a direct target of ATF5 and CEBPD [29,120]. Both survival proteins were downregulated in GBM (T98G, U87MG) and other cancer lines in response to CP-DN-ATF5 [59]. BCL2 and MCL1 proteins were also significantly reduced in T98G and several non-GBM lines after Bpep or Dpep treatment [168]. Darvishi et al. [169] reported that ST101 downregulates *BCL2* mRNA levels, though the effect on protein levels was unclear from the presented data. In the study of Karpel-Massler et al. [59], another pro-survival BCL2 family member, BCL-XL, also showed reduced expression in U87mg and T98G cells in response to CP-DN-ATF5.

Survivin (a product of the BIRC5 gene) is an additional important anti-apoptotic protein that was found responsive to the peptides. It is highly expressed by brain and other cancer cells, but not by most normal cells [171]. Multiple studies have shown that survivin inhibition, loss or downregulation triggers apoptosis of malignant cells, including GBM and medulloblastoma [172,173], and it has also been described as a prognostic indicator for GBM [174]. It has therefore been recognized as an important therapeutic target in brain and other cancers [171,173]. In this context, Sun et al. [175] found that CP-DN-ATF5 rapidly depletes survivin mRNA and protein in T98G, U87, LN229 and GBM12 GBM cells as well as in multiple cancer cell lines of various origins. This effect was at least partially due to destabilization and proteasomal degradation of the protein. Interestingly, survivin over-expression was unable to overcome the capacity of CP-DN-ATF5 to promote apoptosis of T98G, LN229, GBM12 and U251 as well as other non GBM lines. This observation supported the conclusion that while survivin depletion is sufficient to trigger cell death, CP-DN-ATF5 bears additional activities that trigger apoptosis even in the presence of survivin. Beyond CP-DN-ATF5, Zhou et al. [168] showed that survivin protein levels are profoundly depleted in T98G and additional cancer lines after exposure to Dpep and Bpep, while Darvishi et al. [169] reported that ST101 downregulates BIRC5/survivin mRNA and protein in U251 GBM cells and, like CP-DN-ATF5, appears to promote proteasomal degradation of the protein.

In addition to downregulating anti-apoptotic proteins, another important action of the peptides in promoting cell death is upregulation of the pro-apoptotic gene *BMF* (BCL2-modifying factor). In light of a pilot study that found upregulation of *BMF* mRNA in T98G cells after CP-DN-ATF5 treatment, Zhou et al. [168] examined regulation of the gene in response to Bpep and Dpep in T98G and several non-glioma tumor lines. Both peptides produced a significant elevation of message levels in all cases. Importantly, downregulation of BMF with siRNA produced a significant decrease in apoptotic death triggered by Dpep and Bpep, thus implicating it as a required player in the mechanisms of action of the two peptides.

In aggregate, the above findings support the conclusion that a major mechanism by which CP-DN-ATF5, Dpep, Bpep and ST101 affect brain and other cancers is by triggering their apoptotic death. This occurs at least in part by downregulating several major pro-survival proteins (BCL2, MCL1, Survivin) and upregulating a pro-apoptotic protein (BMF). As will be discussed below, such observations have implications for potential partners with which the peptides might be combined to enhance efficacy.

### 11.3. Consideration of Proliferation and Cell Cycle

An important, but not fully resolved issue is whether CP-DN-ATF5, Dpep, Bpep and ST101 have cytostatic as well as cytotoxic activities. In the case of ST101, it was reported that U251 glioma cultures synchronized by thymidine block showed a significant increase in cells in the G1 phase after exposure to the peptide, suggesting a block in the cycle at G1/S [169]. However, it was unclear whether this was persistent or how the onset of apoptosis might have affected the results. A RNAseq study [169] in U251 GBM and two other non-glioma cell lines also found that ST101 reduces transcripts encoding several cell cycle-related proteins, which could potentially affect the capacity for proliferation, but could also contribute to causing apoptosis. All in all, it appears that further work is warranted to clarify whether the peptides have cytostatic as well as cytotoxic activity, especially with regard to understanding how this might affect the efficacy of their treatment of brain and other tumors in a clinical setting.

A related issue is whether the apoptotic activity of the various cell-penetrating peptides requires that cancer cells be actively in the cell cycle. This is of particular relevance to the question of whether the peptides can kill cancer cells that are in a senescent or growth-arrested state. There is growing evidence that senescent tumor cells, often generated in response to therapies contribute to recurrence of GBM and other malignancies [176,177]. To address this for Bpep and Dpep, Zhou et al. [168] employed a senescence model in which T98G and three non-glioma cell lines were treated with 100–200 nM doxorubicin for 24 hr, which caused them to accumulate in G2/M and to show little or no proliferation for the next 6 days. During that time, the doxorubicin-treated and control proliferating cells were exposed to Dpep or Bpep. In either case, the dose responses for dividing and non-dividing cultures were essentially the same. Such data suggest that at least Bpep and Dpep can potently induce the death of tumor cells that are in a non-proliferating senescent state. Considering that CEBPB has been described as a mediator of oncogene-promoted cellular senescence [178], it remains to be determined whether or not the effects of the peptides on senescent cell survival involve leaving this state prior to undergoing death.

### 11.4. Proximal Transcriptional Actions

It is now well appreciated that altering the activity of even a single transcription factor leads to widespread changes in gene expression patterns and cellular behavior. As reviewed here, perturbation of ATF5/CEBPB/CEBPD affects a variety of properties of brain and other tumor cells such as their survival, growth, migration, mesenchymal transition, and response to therapeutics. This is likely largely ascribable to direct and reverberating changes in transcription. At present, our understanding of such transcriptional effects is incomplete, and probably context dependent. As a step forward, Darvishi et al. [169] presented the results of RNAseq analysis of U251 GBM, A549 (lung cancer) and MCF7 (breast cancer) tumor cell lines that were treated with or without ST101 (concentration not given) for 24 h. Strikingly, there was a large difference in number of differentially expressed genes between the lines (2454, 1443 and 116, respectively), suggesting context-specific responses. GSEA analysis for A549 cells identified alterations in cell cycle and transcription factor (E2F, RUNX and MYC) networks. qPCR studies on all three lines showed variable, dose-dependent (2.5–10 µM; 24 h) decreases in expression of survival factors *BCL2*, *BIRC3*, and *BIRC5*; cell cycle genes *CCNB1*, *CCNA2* and *CDK1*; and ID family genes *ID1*, *ID2* and *ID3*. Western blotting confirmed reduction in most of the corresponding proteins in U251 cells. The observations regarding downregulation of BCL2 and BIRC5/survivin are consistent with previously reported effects of CP-DN-ATF5, Dpep and Bpep on these proteins [59,168,175]. In future, it will be important to extend such analyses to additional cell lines and times for each of the four existing peptides and to tease out which of the many changes observed drive tumor cell responses such as apoptosis. It is anticipated that a better understanding of the proximal transcriptional events triggered by cell-penetrating peptides that target ATF5, CEBPB and CEBPD will better define their mechanisms of action, which in turn will inform their best use in therapies for treatment of brain and other cancers.

### 11.5. Potential Interference with Additional Transcription Factors

Although CP-DN-ATF5, Dpep, Bpep and ST101 have been presently characterized with respect to their interactions with ATF5, CEBPB and CEBPD, there remains the possibility that the peptides also act in part by directly or indirectly interfering with the activities of additional transcription factors. For example, there is evidence [179,180,181,182] that CEBPB directly associates with bZIP protein ATF4, a key regulator of cellular responses to various types of stress that can either protect cancer cells or promote their demise, depending on circumstances [183]. Multiple studies have described roles for ATF4 in driving GBM [184,185,186]. There is also evidence for interaction of CEBPB with leucine zipper protein CHOP (product of the *DDIT3* gene) under conditions of mitochondrial stress [187]. Additionally, it is reported that CEBPB heterodimerizes with the bZIP family member CEBPG [188,189]. CEBPG is a widely expressed protein with described roles in promoting cell proliferation and as a fundamental mediator of the integrated stress response, and has been identified as a probable oncogene in a variety of cancers [190]. Based on the current literature, of the cell-penetrating peptides discussed here, Bpep seems the most likely to interact directly with ATF4, CHOP and CEBPG. Consistent with this, Darvishi et al. [169] provided evidence that ST101, which appears to act more as a DN-ATF5 rather than as a DN-CEBPB, does not associate with CEBPG. On the other hand, ST101 as well as CP-DN-ATF5 and Dpep may indirectly interfere with some functions of ATF4, CHOP and CEBPG by depriving them of access to CEBPB. On these bases, it appears that additional studies are warranted to better define targets of each of the peptides.

## 12. Combination Therapies Employing CP-DN-ATF5, Bpep, Dpep and ST101

Our current and growing understanding of the mechanisms by which the peptides affect the properties of tumor cells has the important potential to inform us how they may be advantageously paired with other anti-cancer treatments in combination therapies. There are many potential advantages of using combination treatments rather than monotherapies for cancer treatment, and findings thus far indicate that this is so for cell-penetrating ATF5, CEBPB and CEBPD decoy peptides [59,168,169]. One appealing aspect to using such peptides in combination therapies is that they themselves appear to have few if any side effects in vivo. Thus, if they have synergistic or even additive activity with other treatments that have toxic side effects, the combination may permit a lowering of the dose of the partner treatment to less- or non-toxic levels while maintaining efficacy. In addition, because the peptides target transcription factors, they have the potential to act orthogonally with treatments directed at other modalities such as kinases. Moreover, while CP-DN-ATF5, BPEP, DPEP and ST101 all show significant anti-tumor activity in in vivo models, for the most part, they appear to slow the growth of tumors rather than eradicate them. In many cases, the growth curves of treated tumors show an initially strong inhibition of growth followed by a slow increase in tumor size over time, suggesting the onset of resistance to peptide treatment. Identification of appropriate combination treatments involving the peptides may provide a more potent and durable response as well as a means to overcome resistance.

### 12.1. BH3-Mimetics

Anti-apoptotic BCL2 family members such as BCL2, MCL1 and BCLXL play major roles in supporting the survival and treatment resistance of gliomas and other cancers [191,192,193]. BH3-mimetics are a series of small molecules that have been developed to exploit the binding specificities of anti-apoptotic BCL2 proteins in order to antagonize them and to promote the apoptotic death of cancer cells [194]. Such mimetics are in clinical use for leukemias and have been advocated for treatment of gliomas based on their sensitivities to these drugs in preclinical studies [192,195]. In this regard, it is relevant that “biomimetic nanoparticles” have been developed to circumvent potential limitations for passage of BH3-mimetics through the blood–brain barrier for glioma treatment [196].

In view of the effects of CP-DN-ATF5 on anti-apoptotic protein BCL2, Karpel-Massler et al. [59] assessed the combination of the peptide with ABT263, a BH3-mimetic BCL2/BCLXL antagonist [197]. This showed a synergistic effect on survival of T98G cells and enhanced apoptosis of LN229, SF188 (pediatric), NCH644 (glioma stem-like) and GBM12 GBM cultures as well as a number of non-glioma cancer lines. Moreover, the combination showed apparent synergistic downregulation of BCL2, MCL1 and BCLXL protein levels in T98G cells. When tested in heterotopic U251 GBM and colorectal cancer xenograft models, the combination of CP-DN-ATF5 with ABT263 significantly reduced tumor growth compared with either agent alone and did not show evident side effects. In the case of U251 GBM cells, the combination produced tumor regression over the 40 day course of the study. Although there are no published data on the combination of BH3-mimetics with Dpep, Bpep or ST101, given the apoptotic actions of these peptides on survival/death proteins, it appears likely that these too will act at least additively with BH3-mimetic agents.

### 12.2. TRAIL

In addition to susceptibility to cell death induced via the “intrinsic” apoptotic pathway, glioma and many other cancer cells can be induced to die by activation of the “extrinsic” pathway by ligands including TRAIL (tumor necrosis factor (TNF)-related apoptosis-inducing ligand). TRAIL has been identified as a potentially attractive cancer treatment based on its selective induction of apoptosis in transformed, but not normal cells [198]. Nevertheless, many cancer types, including glioblastoma, show resistance to TRAIL alone, and therefore, it appears that its most effective use is in combination therapies [199]. In this context, Karpel-Massler et al. [59] found that CP-DN-ATF5 sensitized T98G and LN229 GBM cells as well as a breast tumor line to TRAIL-induced apoptosis. Interestingly, the combination showed enhanced downregulation of BCL2 and MCL1 proteins and at least part of the mechanism by which the sensitization occurred was due to MCL1 depletion. These findings thus raise the possibility of a combination therapy of TRAIL with the peptides described here.

### 12.3. Radiotherapy

Radiotherapy is part of the standard of care for GBM and other cancers and significantly, ATF5, CEBPB and CEBPD have been associated with cancer cell radiation resistance [127,200,201]. Zhou et al. [168] assessed the combinations of Dpep and Bpep with gamma radiation in T98G GBM and HCT116 colon cancer cells. The combinations produced additive to synergistic apoptotic effects, with the latter being more pronounced for a paradigm in which radiation was given 24 h prior to Dpep/Bpep treatment, as opposed to both applied at approximately the same time. These findings thus support further studies on use of peptides targeting ATF5 and/or CEBPB and CEBPD in conjunction with radiotherapy.

### 12.4. Temozolomide (TMZ)

TMZ is an additional standard of care therapeutic for gliomas. Using a subcutaneous U251 GBM xenograft model, Darvishi et al. [169] delivered ST101 at a subtherapeutic dose of 10 mg/kg 3x/week for 3 weeks along with orally administered 100 mg/kg TMZ three times per week for 1 week. While neither treatment alone affected tumor size, the combination significantly slowed tumor growth. In a similar xenograft study with T98G cells, the combination slowed tumor growth significantly better than ST101 alone.

### 12.5. Paclitaxel

The microtubule stabilizing drug paclitaxel is used for clinical treatment of breast and other cancers, and preclinical studies suggest that with appropriate delivery means, it is a good candidate for treatment of gliomas [202,203]. Zhou et al. [168] found that the combination of paclitaxel with Dpep or Bpep showed synergistic efficacy on T98G cells and on two breast cancer lines. Dpep and Bpep also showed full potency on T98G cells that were selected for resistance to paclitaxel. These observations are consistent with findings that ATF5, CEBPB and CEBPD regulate the paclitaxel responsiveness of cancer cells [98,121,143] and support the possible combination of the peptides with this drug.

### 12.6. Chloroquine

The lysosomal inhibitor chloroquine is another drug that has been suggested as a potential therapy for gliomas and other cancers [204]. Moreover, it has been reported to enhance the release of cell-penetrating peptides from endosomes [205]. When tested on T98G and breast cancer cells, chloroquine was found to act synergistically with Dpep and Bpep [168].

### 12.7. Doxorubicin

The anthracycline doxorubicin is widely used to treat various cancers [206] and shows promise for glioma treatment in preclinical models such as those employing nanoparticle delivery to pass the BBB [207]. Zhou et al. [168] showed that the combinations of doxorubicin (50 nM) with Dpep or Bpep provided near-additive activity when assessed on T98G cells and three other non-glioma cell lines.

Taken together, the data reviewed here support the potential application of cell-penetrating peptides targeting ATF5 and/or CEBPB and CEBPD as part of combinations with therapies that are already in clinical use or development. This is particularly appealing for agents with potentially severe side effects and narrow therapeutic windows such as many of those discussed above. It will be important in future to assess such combinations in additional pre-clinical models. Moreover, there is likely significant value in designing additional combinations that are rationally based on insights provided by deeper exploration of the mechanisms by which the peptides act and by which tumor cells may develop resistance to them.

## 13. Clinical Trial with ST101 for Recurrent GBM

As reported by ClinicalTrials.gov, “A Phase 1–2 Study of ST101 in Patients With Advanced Solid Tumors” (NCT04478279) was posted in July 2020. The trials included GBM among the solid tumors listed. An abstract of a presentation at the November 2022 meeting of the Society for Neuro-Oncology entitled “Early signal of activity from a phase 2 study of ST101, a first-in-class peptide antagonist of CCAAT/enhancer-binding proteinβ (C/EBPβ), in recurrent glioblastoma (GBM)“ provided the first glimpse of the outcome for GBM [208]. The abstract reported that the study enrolled adult patients with tumors that had recurred or progressed after one standard treatment regimen. ST101 treatment was 500 mg delivered weekly by IV. Of the seven patients that reached the first on-study assessment at 18 weeks, six progressed and one had a confirmed partial response as determined by mRANO criteria. It was also reported that “ST101 has a favorable safety profile with minor infusion related reactions being the most common adverse event.” While the study is at an early phase, the apparent safety of the drug and its potential efficacy in at least one patient are presently encouraging.

## 14. Conclusions and Perspectives—What Is Next?

We have described the existing evidence that ATF5, CEBPB and CEBPD are major targets for treatment of brain and other cancers, how the molecular properties of these transcription factors have provided a strategy to interfere with their activities, and how the use of a cell-penetrating sequence has enabled successful design of peptide drugs that show significant promise for clinical use. What then lies ahead in both the near and far terms? First, in the near term, the ongoing ST101 trial should provide valuable findings that will inform us about the efficacy and safety of such peptides in patients as well as their potential best use in the clinic. Second, the advancement of ST101 to phase 1–2 trials and an early promising indication bodes well for development of additional cell-penetrating peptides that target ATF5 and/or CEBPB and CEBPD. As noted, peptides such as Bpep and Dpep that target ATF5 directly as well as CEBPB and CEBPD may have potential advantages over ones designed to target CEBPB alone. Going forward, there is room to improve the efficacy of such peptides, for example, by enhancing their interactions with their targets and by increasing their in vivo stabilities. For example, recent major advances in the modeling of protein structures [8,9,209] and docking of peptide ligands with their protein targets [167,210] will likely enable further optimization of sequences for target binding and specificity. The use of D-amino acids such as successfully done with ST101 [169] may also prove to increase stability and therefore efficacy. As discussed above, a third area that merits further near-term investigation is the identification and development of combination therapies that include cell-penetrating peptides targeting ATF5 and/or CEBPB and CEBPD. Fourth, as noted, there is a need for additional studies on the mechanisms by which the peptides act on tumor cells, particularly regarding the underlying perturbations of transcription. Fifth, the available data suggest unsurprisingly that cancer cells can develop resistance to the cell-penetrating peptides described here. Therefore, it will be important to investigate the mechanisms by which this occurs, which will in turn inform regarding potential means to avoid and reverse such resistance. Sixth, the bulk of studies reviewed here regarding ATF5, CEBPB and CEBPD in brain tumors mostly concern GBM. There is a need to systematically assess the effects of the peptides on other brain-intrinsic tumors such as medulloblastoma and low-grade gliomas, as well as on tumors that metastasize to the brain. Seventh, given that the systemically delivered peptides are anticipated to have access to all cells in the body in addition to malignancies, they may have effects on the tumor microenvironment and/or immune system that are relevant to their clinical efficacy. Thus, future work should be invested in evaluating such potential actions.

Several longer-term possibilities also merit consideration. The strategy of using cell-penetrating decoy peptides that exploit strong and selective protein–protein interactions is by no means limited to the transcription factors discussed here and has been effectively used elsewhere [211,212]. This approach is therefore highly amenable to targeting additional proteins that play key roles in brain and other tumor cells. A variety of such proteins have been and continue to be identified by contemporary screening methods. At a different level, once targets for brain and other cancers such as ATF5, CEBPB and CEBPD are identified, there is the possibility to drug them with small molecules. Though transcription factors had at one time been considered as “undruggable”, the ongoing rapid advancements in computational modeling of protein structure indicate a way forward to identify suitable molecular surfaces for the design of interacting and inhibitory small molecule drugs.

## Figures and Tables

**Figure 1 cells-12-00581-f001:**
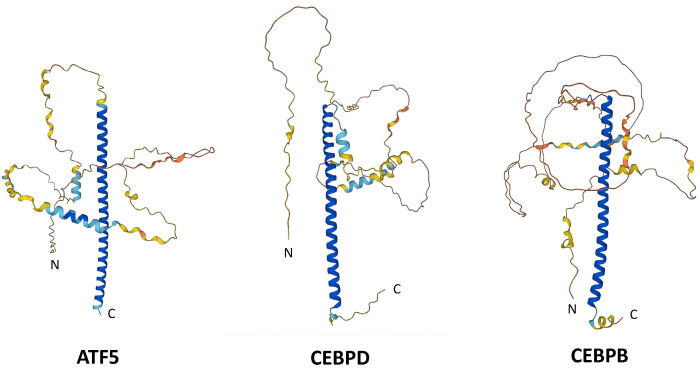
Structures of human ATF5, CEBPD and CEBPD as predicted by AlphaFold [8,9]. The N- and C-termini are indicated. The bZIP domains appear as the vertical coils in blue.

**Figure 2 cells-12-00581-f002:**
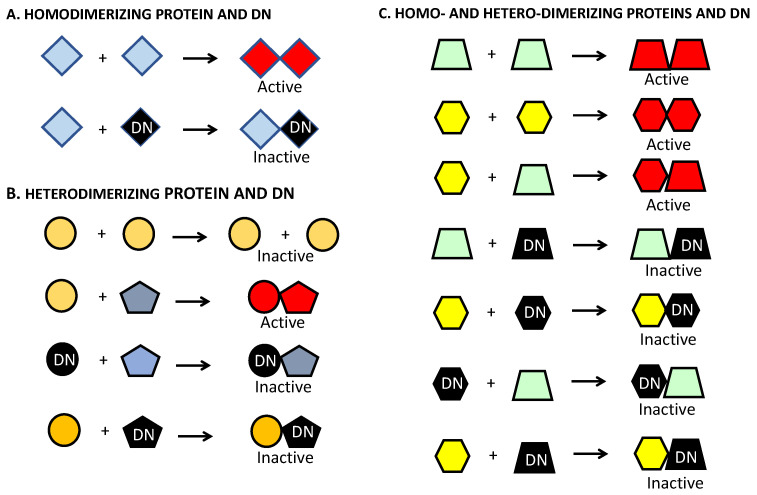
Examples of interference with transcription factor activity by dominant negatives (DN). (**A**). A factor that is activated by homodimerization. Blue symbols depict inactive wild-type monomer; black symbols, DN mutant form of the same factor; red symbols, activated wild-type homodimer. The wild-type:DN heterodimer is inactive. (**B**). A factor that does not homodimerize and is activated by heterodimerization. Orange symbols depict inactive wild-type monomer; blue-gray symbols, wild-type inactive heterodimerizing partner; black symbols, DN mutant forms of each factor; red symbols, activated wild-type heterodimer. Heterodimers formed by either of the wild-types with DN forms of heterodimerization partners are inactive. DN-ATF5 appears to conform to this model. (**C**). Two factors that are activated by homodimerizing and by heterodimerizing with one another. Green and yellow symbols depict inactive wild-type monomers; black symbols, DN mutant forms of each factor; red symbols, activated wild-type homo- and heterodimers. Heterodimers formed by each wild-type factor and its corresponding DN are inactive as are heterodimers formed by each wild-type factor and the DN of its wild-type heterodimerization partner. Dpep and Bpep appear to follow this mode.

**Figure 3 cells-12-00581-f003:**
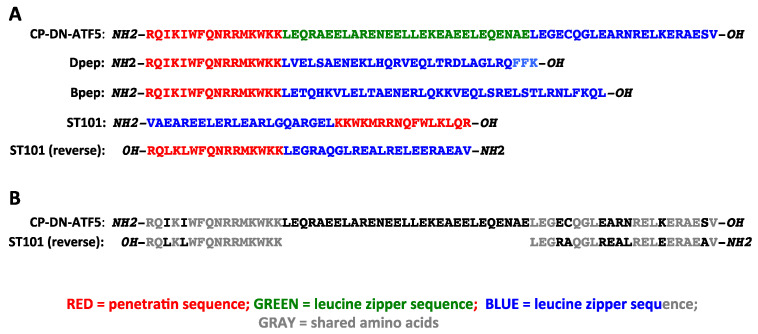
Comparison of the sequences of CP-DN-ATF5, Bpep, Dpep and ST101. (**A**). Sequences of each of the peptides. The various domains of each peptide are denoted in the color schemes given in the figure. Note that ST101 is composed of D-amino acids. The reverse sequence is also shown. (**B**). Comparison of the reverse sequence of ST101 with CP-DN-ATF5.

**Figure 4 cells-12-00581-f004:**
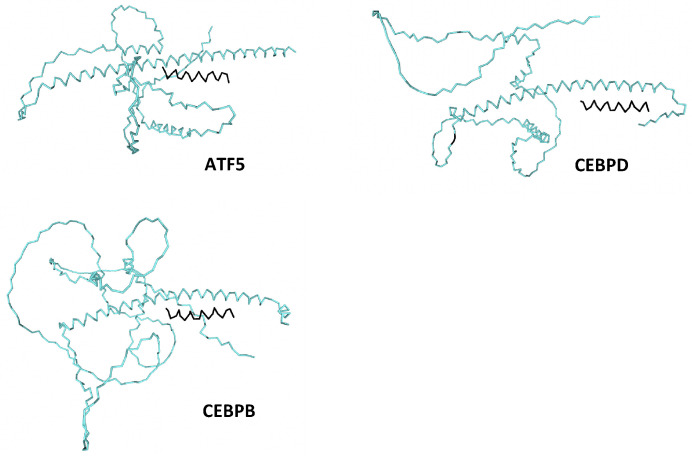
Computed sites for association of a portion of the CEBPD leucine zipper (21 amino acids, black) with ATF5, CEBPB and CEBPD. The molecular structures of ATF5, CEBPB and CEBPD were generated by the AlphaFold protein structure database [8,9]. The structure of the CEBPD leucine zipper peptide was computed by the PatchMAN protocol as were the docking sites for the peptide with ATF5, CEBPB and CEBPD [167]. The ensuing models were rendered using UGENE. In each case, the PatchMAN model 1 is shown.

**Table 1 cells-12-00581-t001:** Target specificities of reported cell-penetrating peptides with respect to ATF5, CEBPB and CEBPD. Data are presently unavailable for association of ST101 with CEBPD, but this appears likely given the similarity of its leucine zipper sequence to that of CP-DN-ATF5.

Peptide	ATF5	CEBPB	CEBPD
CP-DN-ATF5	−	+	+
Dpep	+	+	+
Bpep	+	+	+
ST101	−	+	+?

## Data Availability

Not applicable.

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
