# Peer review of "Targeting Transcription Factors ATF5, CEBPB and CEBPD with Cell-Penetrating Peptides to Treat Brain and Other Cancers"

_cells, 2023, doi:10.3390/cells12040581_

Round 1

Reviewer 1 Report

In this review the authors focus their attention on the leucine zipper transcription factors ATF5, CE- 13 BPB, and CEBPD as targets for brain and other malignancies.

They cover extensively up-to-date literature on the topic focusing on in vitro and in vivo publications regarding the impact of these transcription factors in the development  of the glioblastoma and other brain tumors with low grade. Additionally, they correlate the upregulation of this factors as a marker for the glioblastoma transition from a proneuronal profile to mesenchymal.

Overall this review is well written and covers the all necessary literature on the topic. It is well organized and suitable for a broad audience of researchers in the field of brain cancer.

I, consequently, do not have any major points in this revision of the manuscript and I support the publication of it.

Kind regards,

Author Response

Thank you for your review and supportive comments.

Reviewer 2 Report

In study, the author identified the basic leucine zipper transcription factors ATF5, CEBPB, and CEBPD as targets for brain and other malignancies. This study first introduced the structure and functions of ATF5 on multiple cancer cells, and then discussed association of DN-ATF with CEBPB and CEBPD.  The author also compared peptides and ST101 independently or in vivo and in clinical trials for solid tumors especially for GBM. Overall, this study pave way for short- and long-term directions for creating new generations of drugs targeting ATF5, CEBPB, CEBPD, and other transcription factors for treating brain and other malignancies and some questions that I think needed to be addressed before publication.
1. The author mainly focused on the evaluation of ATF5, CEBPB, and CEBPD on brain cancer and relevant descriptions of these targets on other malignancies need further discussion.

2. The author cited the pulldown-western blot results and confirmed that ATF5 can interact with CEBPB and CEBPD. The part was suggested to add more evidence to demonstrate the regulation between these genes by other means, such as chip-sequencing. The authors also need supplement corresponding literatures of CEBPB and CEBPD by ATF5 in other cell lines and potential mechanism.

3. This study only describe functional behavior changs of these tumor cell lines under epigenetic regulation by ATF5, CEBPB, and CEBPD, and tumor microenvironment regulation by these genes should need further elaboration.

4. The author only introduced clinical trial concerning ST101 for recurrent GBM treatment. Are there other clinical trials concerning single ST101, CP-DN-ATF5, Bpep and Dpep? Or are there the combination of two or more factors in clinical trials concerning GBM and other tumors?

5. Studies related to the treatment of metastatic malignant tumor of brain with ST101, CP-DN-ATF5, Bpep and Dpep, or intrinsic mechanism regulated by ATF5, CEBPB, and CEBPD should be included.

6. This study has few tables and figures describing interactions between ATF5 and its downstream proteins. Subsequent discussion on the therapies targeting these genes also need to be listed in table.

Author Response

REVIEWER 2

We thank the reviewer for their comments and suggestions. Our responses are given below in italics.

In study, the author identified the basic leucine zipper transcription factors ATF5, CEBPB, and CEBPD as targets for brain and other malignancies. This study first introduced the structure and functions of ATF5 on multiple cancer cells, and then discussed association of DN-ATF with CEBPB and CEBPD.  The author also compared peptides and ST101 independently or in vivo and in clinical trials for solid tumors especially for GBM. Overall, this study pave way for short- and long-term directions for creating new generations of drugs targeting ATF5, CEBPB, CEBPD, and other transcription factors for treating brain and other malignancies and some questions that I think needed to be addressed before publication.

  1. The author mainly focused on the evaluation of ATF5, CEBPB, and CEBPD on brain cancer and relevant descriptions of these targets on other malignancies need further discussion.

Response: We agree that we have focused on brain cancers. This the specific subject of the special issue to which we were invited to submit, and we have tried to make it clear that our intention was to focus on these types of tumors. We have however included numerous references to the roles of ATF5, CEBPB and CEBPD in other cancer types in section 8 of the review and have described and referenced the effects of the various peptides on other types of malignancies throughout the text. We agree that it would be useful to review in more detail the roles of ATF5, CEBPB and CEBPD in non-brain malignancies, but we do not believe that this is the appropriate venue in which to do so.

  1. The author cited the pulldown-western blot results and confirmed that ATF5 can interact with CEBPB and CEBPD. The part was suggested to add more evidence to demonstrate the regulation between these genes by other means, such as chip-sequencing. The authors also need supplement corresponding literatures of CEBPB and CEBPD by ATF5 in other cell lines and potential mechanism.

Response: We agree that it would be interesting to have data showing that the various transcription factors bind the same sites in the genome. However, we are unaware of such systematic data being available and given that this is a review of published literature, we could not include such data. Also, the key to the action of the peptides described here is that they interact physically with CEBPB, CEBPD and/or ATF5, thus blocking their interaction with DNA. Regarding the “other cell lines” referred to above, we believe that we have amply referenced such work where relevant, especially with respect to mechanism. If there are other specific references that are missing, we would be glad to know about them.

  1. This study only describe functional behavior changs of these tumor cell lines under epigenetic regulation by ATF5, CEBPB, and CEBPD, and tumor microenvironment regulation by these genes should need further elaboration.

Response: We have not considered epigenetic regulation by ATF5, CEBPB and CEBPD and are not aware of literature on this subject.  We agree that the effects of these factors and corresponding peptides on the tumor microenvironment is an interesting subject, but we are not aware of a coherent literature on this for brain or other malignancies. In response to this point, we have revised the manuscript by adding material starting at line 1156 that stresses the importance of future work to determine the possible effects of the peptides on the microenvironment.

  1. The author only introduced clinical trial concerning ST101 for recurrent GBM treatment. Are there other clinical trials concerning single ST101, CP-DN-ATF5, Bpep and Dpep? Or are there the combination of two or more factors in clinical trials concerning GBM and other tumors?

Response: The recent abstract on the ST101 trial is the only clinical report presently available in the literature that we are aware of.  Our review also notes the description of the intended ST101 trial on various tumor types as it appears in Clincaltrials.gov. There are no monotherapy or combination therapy trials for the other peptides that are currently ongoing or planned. Thus, we have described all clinical work with the peptides that we are aware of.

  1. Studies related to the treatment of metastatic malignant tumor of brain with ST101, CP-DN-ATF5, Bpep and Dpep, or intrinsic mechanism regulated by ATF5, CEBPB, and CEBPD should be included.

Response: As we understand the reviewer’s comments,  they are  referring to tumors that originate outside of the brain and then metastasize to brain. We agree that it would be interesting to assess the presence of ATF5, CEBPB and CEBPD in such tumors and the effects of the various peptides thereon. However, we are unaware of any direct literature on this subject, so cannot include it in this review. Again, if the reviewer is aware of any such studies, we would be glad to know about them.

  1. This study has few tables and figures describing interactions between ATF5 and its downstream proteins. Subsequent discussion on the therapies targeting these genes also need to be listed in table.

Response: We are unclear as to what the reviewer means by “interactions between ATF5 and its downstream proteins”.  Does this refer to protein-protein interactions or to ATF5’s gene targets? In either case, there is only a very limited literature available. For protein interactions, these are included in the body of the review, but given the small number and lack of information on their importance for the actions of the peptides, we do not believe that they merit a table at this time. For direct ATF5 target genes, there is a paucity of information and it is not clear which of such targets would be of therapeutic importance.

Reviewer 3 Report

The manuscript aimed to discuss the role of transcription factors ATF5, CEBPB, and CEBPD to treat brain and other cancers. Overall, the paper is well-written though a lot of information was presented. I think the paper can be accepted in the present form.  

Author Response

Response:  Thank you for your review and supportive comments.